

# Multi-element ducts for ducted wind turbines: A numerical study

Vinit V. Dighe [1], Francesco Avallone [1], Ozer Igra [2], and Gerard van Bussel [1]

[1]Wind Energy Research Group, Faculty of Aerospace Engineering, TU Delft, Delft, The Netherlands
[2]Department of Mechanical Engineering, Ben-Gurion University of Negev, Beersheva, Israel

**Correspondence:** Vinit V. Dighe (V.V.Dighe@tudelft.nl)

**Abstract.** Multi-element ducts are used to improve the aerodynamic performance of ducted wind turbines (DWTs). Steady-state, two-dimensional computational fluid dynamics (CFD) simulations are performed for a multi-element duct geometry, consisting of a duct and a flap; goal is to evaluate the effects on the aerodynamic performance of the radial gap length and the deflection angle of the flap. Solutions from inviscid and viscous flow calculations are compared. It is found that increasing the radial gap length results in an augmentation of the total thrust generated by the DWT, whereas a larger deflection angle has an opposite effect. A reasonable to good agreement is seen between the inviscid and viscous flow calculations, except for multi-element duct configurations characterized by large flap deflection angles. The viscous effects become stronger at large flap deflection angles, and the inviscid calculations are incapable to take into account this phenomenon.

## 1 Introduction

Ducted wind turbines (DWTs) represent an interesting technological solution for increasing the energy extraction with respect to conventional horizontal axis wind turbines (HAWTs) for a given rotor radius and free stream velocity (de Vries , 1979). This solution is particularly suited for urban areas where the radius of the rotor is a constrain and the free stream velocity is low due to presence of buildings. DWTs are constituted of a rotor and a duct; the role of the latter is to increase the flow rate through the rotor relative to a similar rotor operating in the open atmosphere, thereby increasing the generated power. van Bussel (2007), using a one-dimensional momentum theory approach, found that the maximum power coefficient obtained by a DWT can exceed the Betz limit by a factor of 2.5. The best aerodynamic performance for a DWT can be achieved by increasing the duct expansion ratio (Liley and Rainbird , 1956; Foreman et al. , 1978; Loeffler and Vanderbilt , 1978; Foreman and Gilbert , 1984; Samson and Katebi , 2014), i.e. by generating a strong reduction of the static pressure at the duct's exit. As a drawback, for a duct with a large duct outlet to rotor area ratio, flow separation along the duct inner walls might be present (Aranake et al., 2015; Dighe et al. , 2018). Tang et al. (2018) investigated experimentally the effects of variable duct expansion ratios on the aerodynamic performance of DWTs. They found that increasing the expansion ratio over a certain limit results in a power reduction. This was linked to the appearance of flow separation within the inner walls. An alternative approach to improve the aerodynamic performance is to increase the camber of the airfoil, used as a cross-section of the duct, until separation occurs behind the rotor plane (Dighe et al. , 2019). Since, as expected, flow separation has an undesired effect, solutions to prevent flow separation via active boundary layer control techniques have been proposed (Foreman et al. , 1978; Igra , 1977; Abe and Ohya , 2004). However, the performance benefits were limited by the cost of the active system and its installation.



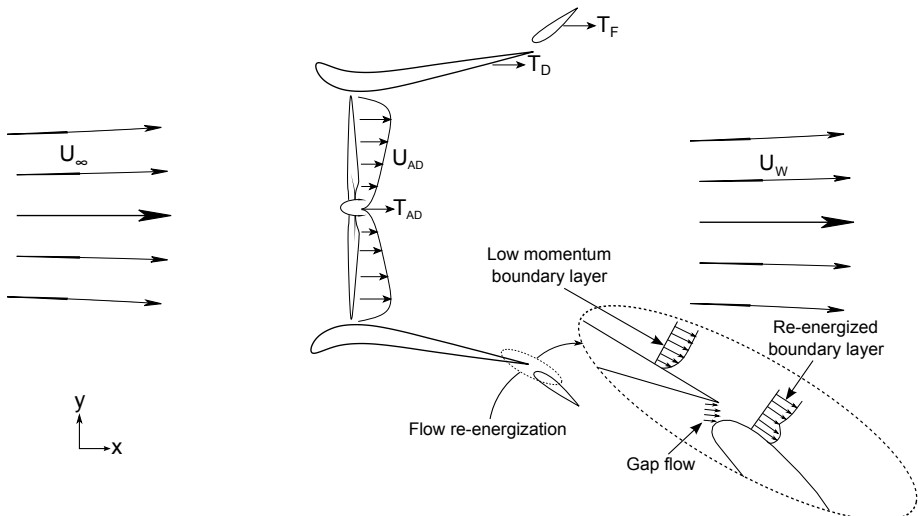

**Figure 1.** Schematic of flow around a multi-element ducted wind turbine.

Another possible solution to improve the aerodynamic performances of DWTs, which has only been explored to a very limited extent, consists of using a duct with a flap (i.e. a multi-element duct). Th flap is realized as a secondary duct with a small chord airfoil cross-section mimicking high-lift devices for aircraft wing (see Figure 1). A first theoretical and experimental analysis of DWTs with a flap was carried out by Foreman et al. (1978) and Igra (1981). The latter found that the addition of a
flap improves the DWTs aerodynamic performance by 25% with respect to a single duct. The flap inhibits flow separation along the inner duct wall and increases the camber of the equivalent airfoil, thus being benefical for the aerodynamic performances of the DWT (Dighe et al. , 2019) . The literature, however, misses out a detailed parametric study that investigates the effect of flap's installation setting, i.e. the radial location and its angle of attack, on the total power generated by a DWT.

Goal of this paper is to conduct a parametric study to investigate the effect of the installation settings of the flap on the
aerodynamic performance of a multi-element DWT. This is performed using Computational Fluid Dynamics (CFD). To this aim, a reference multi-element duct is selected and the rotor is simulated by a uniformly loaded actuator disc (AD) model. The rest of the paper is organized as follows. Section 2 reports the most relevant non-dimensional coefficients adopted for characterizing the multi-element duct-AD model. Section 3 describes the computational settings and parameters with a brief description of the numerical methodology. Section 4 reports the validation of the computational findings. The dependence of
the non-dimensional coefficient on the flap installation settings is discussed in Section 5. Finally, the most relevant results are summarized in the conclusions.





## 2 Multi-element duct-AD flow model

The incompressible flow past a wind turbine is computed substituting the rotor with an AD of infinitesimal width. The AD exerts a uniform thrust force $T_{AD}$ per unit area. Then, the non-dimensional thrust force coefficient is:

$$C_{T,AD} = \frac{T_{AD}}{\frac{1}{2}\rho U_\infty^2 S_{AD}},$$ (1)

5    where $\rho$ is the fluid density, $U_\infty$ is the free stream velocity and $S_{AD}$ is the AD area.

$T_{AD}$ is obtained by forcing an uniform pressure drop across the AD, $T_{AD} = \Delta p \times S_{AD}$. The pressure drop $\Delta p$ is taken from experiments and is given as an input parameter to the numerical simulations. The mean velocity across the AD, $U_{AD}$, is obtained by integrating the difference of the streamwise velocity component across the AD surface $U_x$:

$$U_{AD} = \frac{1}{S_{AD}} \oint_{S_{AD}} U_x dS.$$ (2)

Then, the power coefficient is:

$$C_{P_o} = \frac{P_o}{\frac{1}{2}\rho U_\infty^3 S_{AD}} = \frac{U_{AD}}{U_\infty} C_{T,AD}.$$ (3)

For a multi-element duct-AD configuration, additional thrust forces are exerted by the duct and the flap. Then, the total thrust

15    force $T$ is the vectorial sum of the AD thrust force $T_{AD}$, and of the axial thrust force exerted by the duct $T_D$ and of the flap $T_F$. It can be written as:

$$T = T_{AD} + T_D + T_F = T_{AD} + T_M.$$ (4)

The total thrust coefficient is then defined as:

20    $$C_T = C_{T,AD} + C_{T,M},$$ (5)

where $C_{T,M}$ is the multi-element duct thrust coefficient.





To highlight the relative contribution of the multi-element duct thrust $T_M$ and the AD thrust $T_{AD}$ on to the total thrust $T$, a dimensionless thrust factor $\tau$ is introduced (Bontempo and Manna , 2016):

$$\tau = \frac{T_M}{T_{AD}} = \frac{C_{T,M}}{C_{T,AD}}, \tag{6}$$

so that the total thrust coefficient can be written as:

$$C_T = (1+\tau)C_{T,AD}. \tag{7}$$

Following Bontempo and Manna  (2016), the normalized axial velocity at the AD for a ducted configuration can be also expressed as a function of thrust coefficient:

$$\frac{U_{AD}}{U_\infty} = \frac{1+\tau}{2}\left(1 + \sqrt{1-C_{T,AD}}\right). \tag{8}$$

Using Eqs. (3) and (8), the power coefficient of the multi-element duct-AD model considering $S_{AD}$ as the reference area can be written as:

$$C_P = \frac{1+\tau}{2}\left(1 + \sqrt{1-C_{T,AD}}\right)C_{T,AD}. \tag{9}$$

In Eq. (9), $C_P$ indicates the power coefficient of the multi-element duct-AD model. The above relation is also valid for a simple AD model setting $\tau = 0$:

$$C_{P_o} = \frac{1}{2}\left(1 + \sqrt{1-C_{T,AD}}\right)C_{T,AD}. \tag{10}$$

Equations (9) and (10) can be used to evaluate the contribution of the multi-element duct through a power augmentation parameter $r$:

$$r = \frac{C_P}{C_{P_o}} = 1+\tau = 1 + \frac{C_{T,M}}{C_{T,AD}}. \tag{11}$$





Equation (11) states that $r$ for a multi-element duct-AD model is proportional to the ratio between the multi-element duct thrust coefficient $C_{T,M}$ and the AD thrust coefficient $C_{T,AD}$. Thus, if $\tau > 1$, then a higher power coefficient can be obtained for a multi-element DWT in comparison to a HAWT with the same rotor. The performance coefficients described above can

be evaluated by means of axial momentum theory (AMT) for DWTs (Khamlaj and Rumpfkeil , 2017). However, the AMT cannot be used to estimate the performance of the duct-AD model for a prescribed $C_{T,AD}$ and a given duct geometry. This problem can be solved using numerical solutions based on panel and RANS methods (Bontempo and Manna , 2016; Dighe et al. , 2019).

## 3  Numerical approach

In this section, the numerical methods employed will be briefly described. For an in-depth description, the reader can refer to (Dighe et al. , 2019).

### 3.1  Panel method

A two dimensional potential flow panel method has been used to compute the steady iso-entropic incompressible flow field around the multi-element duct-AD model following the work of de Oliveira et al.  (2016). The governing flow equations are a
simple form of the Euler equations. The AD is represented by a pair of symmetric counter-rotating vortices. The duct and the flap geometries are defined using a distribution of vortex located on panels such to reproduce the desired cross-sectional shape. A uniform distribution of vorticity on the panels is assigned by assuming the Kutta condition. The assumption of uniform vorticity distribution over the panels represents a simplification of real physics and prevents flow separation on the multi-element duct surface, even for larger pressure gradients. The duct and the flap surface discretization is based on the constant
spacing approach. The stream-wise discretization in the near and far wake is non-uniform, with initial panel length equal to 1.0% of duct chord length $c$, just behind the AD, and increasing gradually in length as the wake expansion settles further downstream (see Figure 2).

The panel method is particularly appealing for routine design analysis due to its short execution time. A typical converged panel method solution is obtained in roughly 0.05 hour on a multi-core work-station desktop computer.

### 3.2  RANS method

A commercial CFD solver ANSYS Workbench® has been used for a complete viscous solution of steady incompressible flow around the multi-element duct-AD model. The governing flow equations are the Reynolds-averaged Navier Stokes (RANS) equations. The 2D computational domain is shown in Figure 3, where the distance from the AD location to the domain inlet and outlet are $12c$ and $24c$, respectively. The computational grid consists of quadrilateral cells with minimum $y^+$ value of 1
on the duct walls. Boundary conditions are: a uniform velocity at the inlet, zero gauge static pressure at the outlet, no-slip walls for duct and flap surfaces. A symmetric boundary condition is applied along the center-line axis while a fan boundary



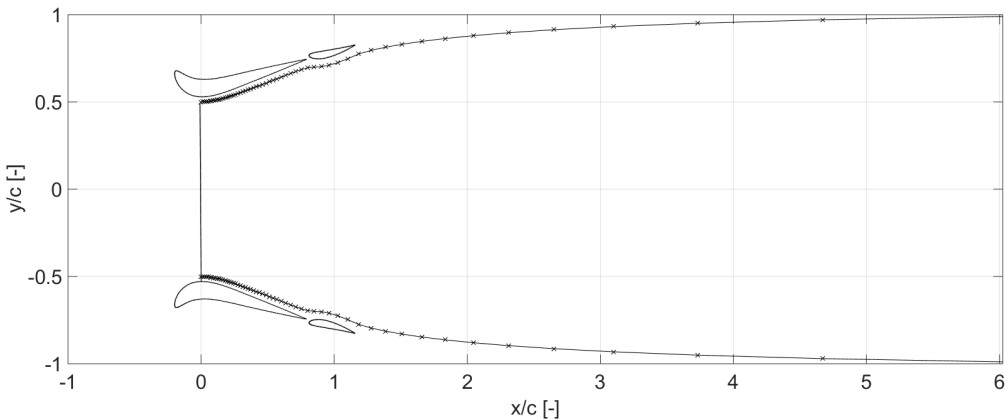

**Figure 2.** Panel distribution along the duct surface and the wake region used for the inviscid panel method calculations.

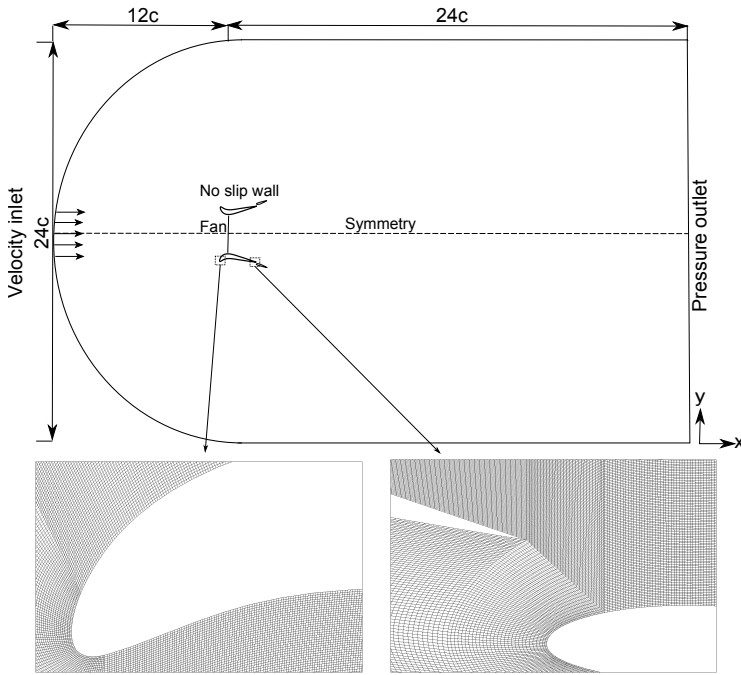

**Figure 3.** Computational domain showing the boundary conditions employed. The length are indicated in terms of duct chord length $c$ (representative, not to scale). The computational grid along the duct's and flap's leading edge are zoomed in to show the C-grid and boundary layer refinement.

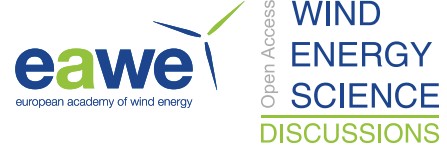

**Table 1.** Grid statistics for grid independence study of the reference case.

| Grid | Number of cells | $C_{T,M}$ |
|--------|-----------------|-----------|
| Coarse | 85890 | 0.1783 |
| Medium | 148380 | 0.18233 |
| Fine | 229300 | 0.18231 |

condition is used for the AD. The k-$\omega$ SST (shear stress transport) model is used as turbulence model. Preliminary investigations showed good agreement with the experiments (Dighe et al. , 2018, 2019). RANS solutions are considerably more reliable and accurate than the panel method solution, but at the expense of computational cost. A typical converged RANS solution with approximately 0.1 million mesh elements is obtained in roughly 0.5 hour on a multi-core work-station desktop computer.

RANS solutions are sensitive to the discretization of the computational domain. For the present computations, a C-grid structured zonal approach is chosen (Figure 3), which proved advantageous in the case of a curved boundary (duct and flap leading edge). The C-shaped loop terminates in the wake region. Grid independence analysis has been carried out using three grid sizes, where the refinement factor in each direction is approximately 1.5. Refinement factor is defined as the rate at which the grid size increases far from the object.

The multi-element duct thrust force coefficient $C_{T,M}$, is taken as reference for the convergence analysis. The results of the grid independence study are shown in Table 1. Convergence is reached for the medium grid, which is then used in the rest of the paper.

## 4   Numerical validation

For validating the numerical methods, experimental data reported by Igra  (1977) were simulated. Igra's experiments were
conducted in the subsonic wind tunnel of the Israel Aircraft Industry; this tunnel has a large test section and it measures 3.6 m × 2.6 m.

Eight different geometries were investigated experimentally, but only two geometries are used for the validation study. The two geometries are: a duct-AD model with $C_{T,AD}$ = 0.434 (named as Model B) and a multi-element duct-AD model with $C_{T,AD}$ = 0.550 (named as Model C (ii) + flap). A schematic of the cross-sections of the two geometries is shown in Figure 4.
The longitudinal cross-section of the duct and of the flap is a NACA 4412 airfoil. The leading edge of both duct geometries are identical. For Model C (ii) + flap, the trailing edge of the duct is radially stretched resulting in a duct expansion ratio $\frac{A_e}{A_{AD}}$ = 1.84, this ratio is 1.71 for model B. The flap chord measures 35% of the duct chord length $c$, and the deflection angle $\theta = 30°$ with respect to the free-stream direction. The experimental dataset consists of: static pressure distribution at different axial and radial positions, and forces generated by the duct and the flap surfaces. During the experiments, the inflow velocity was set at
$U_\infty$ = 32 m/s. Following Igra  (1977), the wall interference and blockage correction can be ignored.

In Figure 5, the power augmentation factor $r$ is plotted as a function of the inflow yaw angle $\alpha$. CFD results, obtained using panel and RANS methods, are compared with the experimental data. A very good agreement between the CFD simulations




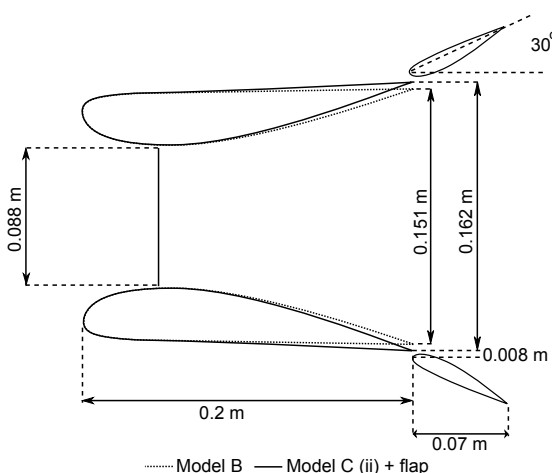

**Figure 4.** A schematic cross-section layout of the three dimensional experimental model used for the numerical validation study.

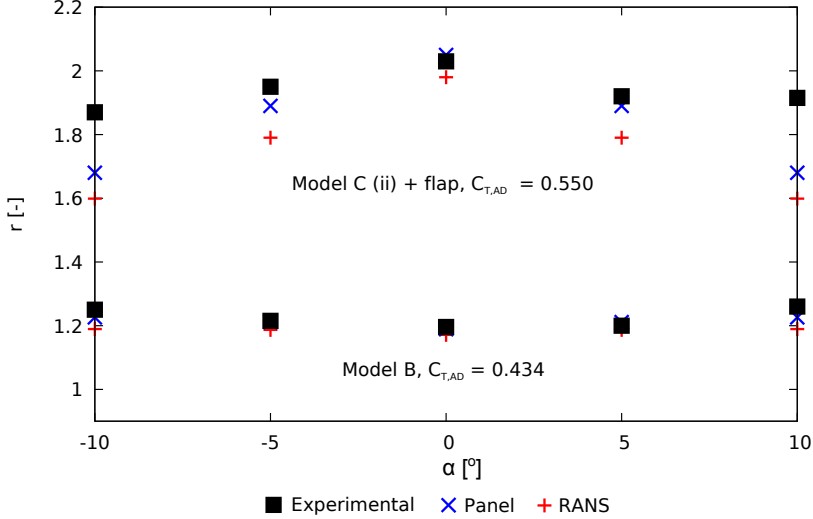

**Figure 5.** Effect of yawed inflow on the power augmentation factor: comparison between experiments (Igra , 1977), panel method and RANS method for different duct configurations.





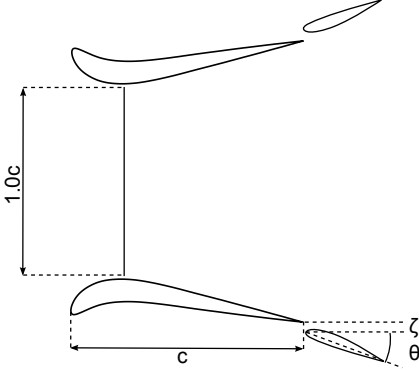

**Figure 6.** A schematic cross-section of the multi-element duct AD model with the variable flap parameters used for the flap installation study.

and the experimental findings is found for Model B. On the other hand, the deviation between the CFD and the experimental findings is larger for Model C (ii) + flap, in particular for $\alpha \neq 0°$. The small discrepancies might be due to three-dimensional effects not accounted in the two dimensional simulations. At non-yawed conditions ($\alpha = 0°$), however, the CFD results agree well with the experimental findings where the maximum deviation of the CFD results from the experimental data is 1.2% and

4.3% for Model B and Model C (ii) + flap respectively. Considering the scope of the current investigation, which does not include the yawed case, the accuracy of the computational codes is considered acceptable.

## 5 Results and discussion

### 5.1 Multi-element duct geometry

In the following sections, the effects of flap installation settings on the aerodynamic performance of the multi-element duct-

AD model are described. The multi-element duct-AD configuration investigated in the present work is shown in Figure 6. The longitudinal cross-section of the duct is a DonQi D5 airfoil; the profile is chosen based on the duct shape parametrization study conducted by the authors (Dighe et al. , 2019). For the DonQi D5 duct an optimal $C_{T,AD} = 0.7$ was obtained. This value is employed throughout the present discussion. A NACA 4412 longitudinal cross-section, measuring $0.35c$, is chosen for the flap following Igra  (1977). The flap installation settings are: the radial gap $\zeta$ and the deflection angle $\theta$. The radial gap $\zeta$, indicated

as percentage of duct chord length $c$, is defined as the distance from the trailing edge of the duct to the leading edge of the flap. A positive value of radial gap ($\zeta > 0$) indicates that leading edge of the flap is positioned below the trailing edge of the duct. A positive deflection angle ($\theta > 0$) corresponds to a downward flap deflection, where the angle is defined relative to the free-stream direction. The axial gap between the trailing edge of the duct to the leading edge of the flap is zero based on the findings of Igra  (1981). In subsections 5.2 and 5.3, the changes in the aerodynamic performance coefficients with respect to

the flap's geometric orientation are quantified.





## 5.2 Duct force coefficient

Contours of the multi-element duct force coefficient $C_{T,M}$ obtained from panel and RANS methods, are shown in Figures 7 and 8, where $C_{T,M}$ as a function of the radial gap $\zeta$ and the deflection angle $\theta$ are reported. The figures show that $C_{T,M}$ increases for larger $\zeta$. Conversely, $C_{T,M}$ decreases with the increasing $\theta$. The differences between results obtained using the panel and RANS methods are smaller than 5% for $\theta \leq 60°$. The maximum $C_{T,M}$ obtained from both the numerical methods lie in the same region, i.e. $\zeta \approx 5$ % and $\theta \approx 10°$. The differences between the two methods increase for $\theta \geq 60°$.

The differences can be explained by looking at the flow field. Contours of non-dimensional axial velocity $\frac{U_x}{U_\infty}$ from both methods are reported in Figures 9 (a)-(f). Results from the panel method are plotted on the left while the ones from RANS on the right. Contours for no-flap configuration are shown in Figures 9 (a) and (b). Two flap settings, in order to explain the aerodynamics behind Figures 7 and 8, are shown: $\zeta = 5$ % and $\theta = 10°$ in Figures 9 (c) and (d) and $\zeta = 5$ % and $\theta = 70°$ in Figures 9 (e) and (f).

Contour plots show a higher velocity at the rotor plane for th configuration with flap in comparison with no-flap configuration. This is due to the additional aerodynamic thrust force generated by the flap. The presence of a radial gap between the duct and the flap accelerates the flow over the flap. This reduces the pressure recovery demands on multi-element duct, thereby reducing flow separation. Obviously, flow separation is seen for RANS contours only. The overall integral contribution of the viscous forces increase the $C_{T,M}$ magnitude in the RANS solutions relative to the panel solutions; a trend that can be clearly observed by comparing Figures 7 and 8. For the flap configuration with $\zeta = 5$ % and $\theta = 70°$, Figure 9 (f), the flow over the flap's inner walls separates completely. The separation along the inner walls of the multi-element duct reduces the $C_{T,M}$, which rapidly becomes large and negative at higher flap deflection angles as seen in Figure 8. For panel solutions, however, the drop in the $C_{T,M}$ magnitude for higher flap deflection angles is gradual (see Figure 7) because viscous effects are neglected (see Figure 9 (e)).

## 5.3 Power augmentation

Figures 10 and 11 represents contours of power augmentation factor $r$, using the panel and RANS solutions respectively, as a function of radial gap $\zeta$ and deflection angle $\theta$. Recall, from Eq. 11, that a $r$ gain for a multi-element duct-AD model can be attained by increasing the $C_{T,M}$ magnitude for a constant $C_{T,AD}$. Evidence of this is provided in Figures 10 and 11, which exhibits the $r$ maximum in the same region of $C_{T,M}$ maximum, as in Figures 7 and 8. Then the maximum power augmentation factor $r = 1.25$ and 1.38, obtained for panel and RANS calculations respectively, corresponds to $\zeta \approx 5$ % and $\theta \approx 10°$.

## 6 Conclusions

In this work, the aerodynamic performance of multi-element DWT is studied using a numerical approach. To this aim, two-dimensional numerical calculations using the panel method and the RANS method are employed. A simplified AD model is used for simulating the rotor. Based on the existing studies conducted by the authors, the multi-element duct geometry




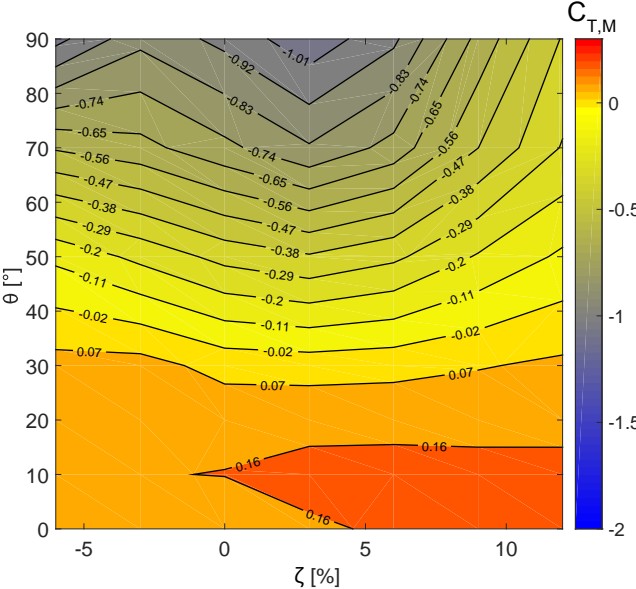

**Figure 7.** Effect of variable radial gap and deflection angle of the flap on the duct thrust force coefficient using panel method.

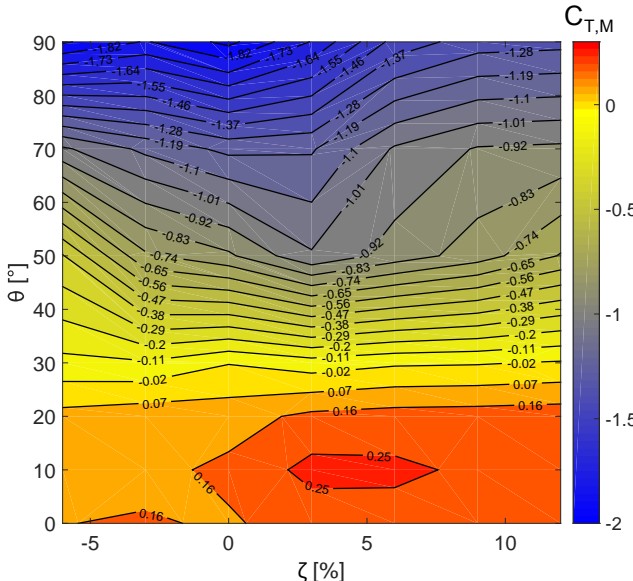

**Figure 8.** Effect of variable radial gap and deflection angle of the flap on the duct thrust force coefficient using RANS method.


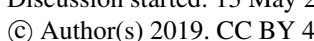

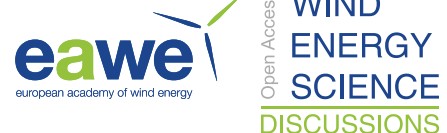

**Figure 9.** Velocity contours colored with normalized free stream velocity obtained using: (a) panel method; no flap, (b) RANS method; no flap, (c) panel method; $\zeta = 5\ \%$ and $\theta = 10°$, (d) RANS method; $\zeta = 5\ \%$ and $\theta = 10°$, (e) panel method; $\zeta = 5\ \%$ and $\theta = 70°$, and (f) RANS method; $\zeta = 5\ \%$ and $\theta = 70°$.




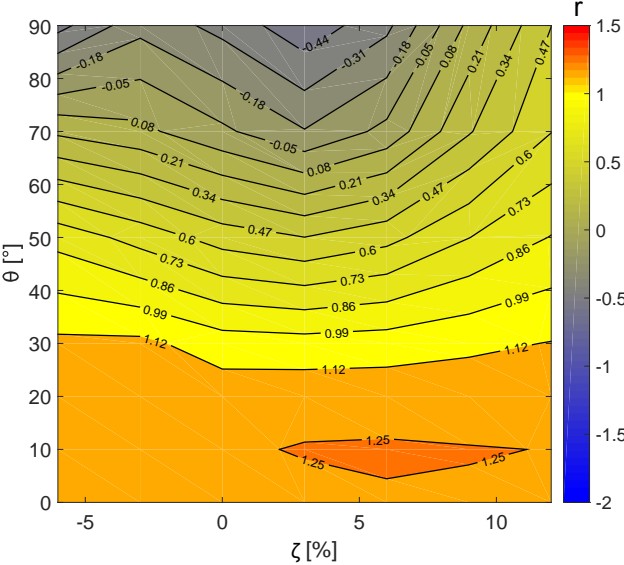

**Figure 10.** Effect of variable radial gap and the flap deflection angle on the power augmentation factor using the panel method.

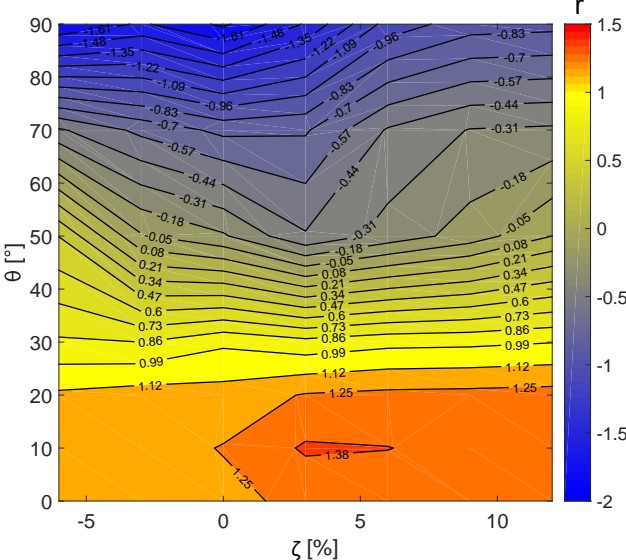

**Figure 11.** Effect of variable radial gap and the flap deflection angle on the power augmentation factor using the RANS method.



consists of a DonQi D5 airfoil and a NACA 4415 airfoil for the duct and the flap cross-sections respectively. To validate the numerical methods, the present simulations are compared with similar experimental data. In order to deepen the design principles of multi-element ducts, the effects of radial gap $\zeta$ and the flap deflection angle $\theta$ on the global performance of DWT are investigated. Clear trends of the multi-element duct thrust force coefficients $C_{T,M}$ and the power augmentation factor $r$ are

5    observed across a range of multi-element duct configurations. An increase in the flap deflection angle $\theta$ result in a decrease in $C_{T,M}$, whereas, increase in the radial gap $\zeta$ shows an increase in $C_{T,M}$. The analysis of flow field shows that flow separation in the multi-element duct inner walls increases for higher values of $\theta$. This phenomenon determines the reduction in $C_{T,M}$, and ultimately the augmentation factor $r$. As expected, the RANS method is more suitable for representing solutions for highly deflected flap configurations. The viscous effects become stronger at higher flap deflection angles, and the panel method is

10   inherently incapable to take account for it. Regarding prediction of near-optimal multi-element-duct configuration, both the numerical methods show good agreement.



*Author contributions.* VVD compiled the literature review, performed the CFD simulations, post-processed the cases and wrote the bulk of the paper. FA contributed in almost all aspects of this study. OI provided the experimental data and participated in structuring of the paper. GVB helped formulate the ideas in regular group discussions.

*Competing interests.* The authors declare that they have no conflict of interest.

5 *Acknowledgements.* The authors would like to acknowledge Dr. Gael de Oliveira, who served as a developer of the panel method used in this paper. The research is supported by STW organization, grant number- 12728.





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
