# Peer review of "Multi-element ducts for ducted wind turbines: A numerical study"

_Wind Energy Science, 2019_

## Short Comment (SC1) · 15 May 2019

Great paper overall!! Glad to see that advanced Research is on-going in this field for small scale wind-energy. I generally agree with the conclusion of the paper. I also have a few questions:

In regards to the experimental data set from a wind tunnel experiment by Igra & Olsen, 1. Is the Reynolds number disclosed for the experimental test data from Igra/Olsen? It appears to be approximately .$\sim 400000 - 440000$ at 32 m/s, based on the model dimensions listed in the paper. Pls. correct if actual Re data is known. What is the Reynolds# (Re)range in the RANS/Panel iterations?

2. The augmentation factor scale for the Igra/Olsen experimental data set shows a rise

in power augmentation (for yawed inflows, Fig. 5, Model B) or 130% power augmentation at +/- 10° inflow compared to 120% power augmentation for 0° in-line flow. Is that the correct interpretation for Model B, Figure 5? The RANS and Panel solutions for Model B show very little change in power augmentation from yawed inflows +/-10° and appears to be consistently 120% power augmentation factor.

3. The optimum design, from the paper's conclusion, is then: the multi-element configuration with an axial gap clearance of 5% of primary duct chord length, positively displaced (below the TE of the primary duct); second element length that is 35% of chord; and deflection angle of less than 10° The second element camber in the paper (NACA 412) obviously has a camber less than or equal to the primary duct aero-foil camber. Any data on performance for thicker camber foils for the 2nd element that is 35% of chord; and for 2nd elements that exceed the aero-foil chord of the primary duct, by, as an example, 50 to 150% of chord.?

Please also note the supplement to this comment:
https://www.wind-energ-sci-discuss.net/wes-2019-21/wes-2019-21-SC1-supplement.pdf

---

## Author Comment (AC1) · 16 May 2019

Thank you for your valuable comments, appreciate it. The following are the responses to your comments:

1. The Reynolds number (based on the duct nozzle diameter) covered in Igra's experiments (late 1970's) ranged between $5 \times 10^4$ < Re < $3 \times 10^5$. For the numerical study (as for panel and RANS iterations), Re $\approx 3.5 \times 10^5$; the value is based on the geometry currently being tested in the wind tunnel experiments.

2. The augmentation factor (for Model B) denoted by $r$ in the paper increases with the increase in yaw angle $\alpha$; this is rightly interpreted by you. The augmentation factor $r$ for different values of $\alpha$ depends on the shape of the duct and the mutual interaction

of the duct and the rotor. The preliminary results highlight the advantage of DWT configuration in urban flows, where the presence of infrastructure/buildings disturbs the flow uniformity resulting in reduced wind speed. A detailed study on characterization of the aerodynamic performance for DWT in yawed flow was beyond the scope of the current article; this is one of the ongoing work and will be published soon. Comparing the experimental data (Model B) with the numerical results (Figure 5) shows a good validation for augmentation factor $r$; deviation $\leq$ 4%. The deviation might be due to three-dimensional effects not accounted in the two dimensional simulations.

3. The numerical study exhibit an optimal configuration for the given geometrical parameters (shape and orientation) of the multi-element duct-AD model. The local maximum, however, will be different for different multi-element geometry and the choice of AD (rotor) loading. The near-optimal region is well captured by both the numerical methods. The panel code (single and multi-element duct configuration) is freely available on contacting the authors.

---

## Referee Comment (RC1) · Anonymous Referee #1 · 13 Jun 2019

The authors performed the optimization of the design of a multi-element diffuser for the improvement of the performance of small size HAWTs. For the purpose, both inviscid and viscous numerical approaches were adopted, which employed an Actuator Disk (AD) methodology for the modelling of the wind-rotor interaction. The robustness of the approach was ensured by validation on available experimental data. Final result of the study is the configuration of the diffuser flaps, in terms of radial gap and deflection angle, which maximizes the overall power coefficient.

The reviewer believes that the topic and the activity are very interesting and worthy of investigation. The paper is clearly coherent with a broader research project, developed by the authors in previous works. It is well presented and the results are clear.

Some specific considerations:

- In the introduction, the presentation of ducted HAWTs technology is not very clear especially as far as their working principle is concerned.

- The numerical approach seems not to be fully adequate for the analysis. Is seems that the authors have used a steady approach for all tested configurations, although this is surely not suitable for high deflection angles because of the intrinsic unsteadiness of the stall phenomenon. The reviewer recommends to verify the validity of the adopted approach by re-analyzing a few selected configurations, especially at high deflection angles, with an unsteady CFD approach.

- The authors state that "The differences between results obtained using the panel and RANS methods are smaller than 5% for $\theta \leq 60°$". Upon examination of Figures 7,8 ,10, 11, however, the discrepancy between the two methods seems to be much higher.

- The reviewer believes that some of the results' comments require a slight revision. In particular, the physical explanation regarding the role of the diffuser radial gap in increasing flow resistance to separation needs to be revised, since it's effect is more related to the re-energization of the flow itself.

- Some errors are present in the paper:

page 1, line 14: Capital "V" is required after the dot

page 2, line 2: "Th" instead of "The"

page 5, line 2: Basing on described DWT theory, it should be "$\tau$>0" and not "$\tau$>1", for an improvement of the power coefficient with respect to the OWT case.

page 10, line 12: "th" instead of "the"

Based on the aforementioned comments, the reviewer does recommend the publication of this paper after the suggested modifications have been applied.

---

## Author Comment (AC2) · 1 Jul 2019

Thank you for your valuable comments; very useful to improve the quality of the paper. The following are the responses to the points highlighted.

1. A more detailed description of the DWT, in terms of working principle, will be presented in the revised manuscript. 2. We totally agree that flow unsteadiness increase, especially for higher deflection angles. The scope of the present study was to compare the two most widely used methods for the study of DWT. You are indeed correct; for higher deflection angles, the viscous interactions play a dominant role in approximating the physics involved. The solutions will differ when unsteady RANS effects are taken into consideration (at the cost of computation time). In the revised manuscript, a

comparison of the same will be presented in the numerical validation section. 3. The differences between the panel and RANS methods stated in terms of percentage will be re-checked. 4. The errors (including typo's) will be rectified.

---

## Referee Comment (RC2) · Anonymous Referee #2 · 4 Jul 2019

The ducted wind turbine concept may be applied to increase the power production of small wind turbines, especially in urban environment where turbine size, rotor protection and noise generation may become crucial constraints. The paper is interesting and provides information on the application of multi ducted elements, focusing on a parametrical analysis related to two geometrical features, with the aim of optimizing the turbine power coefficient. The main issue of the ducted turbine is the definition of the optimized shape of the diffuser acting downstream of the machine, easily prone to separation leading to a dramatic reduction of the effectiveness of the whole design. The numerical methods applied in the paper are suitable to reproduce the actual behavior of the diffuser, but the analysis of the turbine behavior is definitely too simplified and the results may not be really considered as representative of the actual fluid dynamic

performance of the whole system. The presence of a non uniform flow field together with the presence of an unsteady wake shed by the turbine, may dramatically influence the outcome of the paper: readers should be advised and the influence of the mentioned limits should be reported and discussed in the paper. Generally speaking, the machine is always simplified by a uniformly loaded actuator disc: in the reviewer's opinion this approach limits the results validity when applied to an actual case. Moreover, the theoretical 1D analysis performed in chapter 2 defines the dependency of the power augmentation parameter "r" to the change in the thrust coefficient CT (which is probably required and related to the turbine simulation methodology applied in the numerical schemes) but does not relate it to the area ratio of the duct outlet to turbine sections., which is the most important parameter influencing the performance of the diffusing duct and thus the flow rate interesting the turbine and the extra power production.

1)The application of the inviscid method (panel) provides the expected results: quite accurate up to the onset of separation, unable to capture separation for more aggressive diffuser geometries. Which is the contribution to the paper of the panel method? The conclusion was already known at very beginning. 2)The application of a steady CFD scheme does not count for the effects produced by the passing wakes shed by the rotor and by the flow radial non uniformity, both on the flow field inside of the duct and, in particular, on the inner wall. When operating in aggressive geometry, passing wakes can induce unsteady separation (like dynamic stall) on the duct inner wall. Author should advise and comment on this effect in the paper. 3)It is not clear why authors prefer to refer to a 2D symmetrical scheme rather than to an axis symmetrical one, which is definitively the most suitable one to represent a rotating machine as a turbine, even in a simplified and steady simplification. 4) very short description of the "FAN condition" should be reported in the paper for clarity reasons and reader's convenience. 5)The reference Re number is not reported in the paper, not for the validation cases, nor for the application ones. Please provide. 6)With reference to 5.1: a)information about the value of the duct and flap outlet area (possibly rated to the inlet

one) should be reported in this paper for reader's convenience. b)A comparison of the obtained results to the 1D momentum theory applied to ducted wind turbines, at least in terms of maximum expected power augmentation coefficient, should be included in the paper. The basic 1D momentum analysis shows that the optimal (maximum, under Betz hypothesis) power coefficient Cp of a ducted turbine can be calculated as $C_{(p,max)} = 2/3\, K/\sqrt{3}$ where K is the area ratio of duct outlet area to turbine area (actually, the square of the A/AAD value reported in the paper) At the same time the induction coefficient "a" at the optimum power production condition can be calculated as $a\_opt = 1 - K/\sqrt{3}$. By referring to these two simple relations, authors may provide to readers interested to the topic, useful information about the deviation of the reported results with respect to the theoretically expected ones. The optimal case presented in the paper seems to exhibit an area ratio of $1.5^2 = 2.25$ and the optimal ideal power coefficient results 0.87, approximately 1.5 the Betz one of the not-ducted turbine. The velocity at the disc is in average 1.3 V0; the optimal Vd for the area ratio of 2,25 is 1.3 * V0......not really far from theory. Author's should comment about that and report part of this discussion in the paper, if they find it useful. > Pag 1, line 16/19 : "The best aerodynamic performance...........i.e. by generating a strong reduction of the static pressure at duct's exit" : the static pressure at the duct exit is expected to increase to the external value. The pressure reduction is expected at the nozzle throat, downstream of the disc.> > PAG 5 line 2: tau in eq. 11 should be >0 to guarantee r>0, not tau>1 as reported. > Some typos can also be found in the paper (th for the): please check.

---

## Editor Comment (EC1) · Alessandro Bianchini (Editor) · 15 Jul 2019

Dear authors, at your earliest convenience, please post your response to the comments from Reviewer #2. Moreover, I would kindly ask you to give a more detailed point-to-point response to the comments from Reviewer #1, in order to let me evaluate more in detail the modifications attended in the final paper.
* * *

---

## Author Comment (AC3) · 27 Jul 2019

The authors highly appreciate the efforts of the reviewers for their valuable comments; this enabled improving the quality of the manuscript. The authors carried out the following revision to the manuscript:

Reviewer 1

1. In the introduction, the presentation of ducted HAWTs technology is not very clear especially as far as their working principle is concerned.

Additional statement detailing the principle of DWT is added.

2.The reviewer recommends to verify the validity of the adopted approach by re-

analyzing a few selected configurations, especially at high deflection angles, with an unsteady CFD approach.

The authors verified and validated the effects of unsteady flow in the numerical validation section. The authors response to this is that although unsteady simulations increase the level of description of the unsteady flow due to the multi-element duct-AD interaction, the computing cost issued by going from RANS to URANS does not justify the scope of the current study, where the effects of distributed AD loading, wake rotation, divergence and inflow yaw angle are totally ignored.

3.Upon examination of Figures 7,8 ,10, 11, however, the discrepancy between the two methods seems to be much higher.

The authors re-checked and rectified the numbers.

4. Some errors are present in the paper

The authors rectified the errors.

Reviewer 2

1. Which is the contribution to the paper of the panel method?

The panel method, as expected, proved incompetent when tested for 'more aggressive' duct geometries. However, the panel method was able to capture the global trend and the local maximum for the parametric study investigated in $\frac{1}{10}$ the time required for RANS computations.

2. The application of a steady CFD scheme does not count for the effects produced by the passing wakes shed by the rotor and by the flow radial non uniformity, both on the flow field inside of the duct and, in particular, on the inner wall.

The response to this review is identical to the response 2 for Reviewer 1. The authors verified and validated few selected configuration using URANS approach. The authors

found that although unsteady simulations increase the level of description of the unsteady flow due to the multi-element duct-AD interaction, the computing cost issued by going from RANS to URANS does not justify the scope of the current study, where the effects of distributed AD loading, wake rotation, divergence and inflow yaw angle are totally ignored.

3. It is not clear why authors prefer to refer to a 2D symmetrical scheme rather than to an axis symmetrical one, which is definitively the most suitable one to represent a rotating machine as a turbine, even in a simplified and steady simplification.

An axisymmetric domain would be indeed suitable for the RANS computations; it would be computationally efficient. However, panel code is written suitable for a planar configuration, where axisymmetricity is achieved using a symmetric condition along the centre line axis. For reasonable comparison, similar boundary conditions are chosen for RANS computation.

4. very short description of the "FAN condition" should be reported in the paper for clarity reasons and reader's convenience.

The authors added a short description for the clarity of the reader.

5. The reference Re number is not reported in the paper, not for the val- idation cases, nor for the application ones. Please provide.

The authors specified the $Re$ number in the appropriate sections of the manuscript.

A comparison of the obtained results to the 1D momentum theory applied to ducted wind turbines, at least in terms of maximum expected power augmentation coefficient, should be included in the paper. Please provide.

The authors included the comparison of $r_{max}$ calculated using CFD methods with the AMT approach. Moreover, the calculation for the same is presented in the appendix section.

---

## Author Response (AR2)

**Multi-element ducts for ducted wind turbines: A numerical study**

Vinit V. Dighe [1], Francesco Avallone [1], Ozer Igra [2], and Gerard van Bussel [1]

[1]Wind Energy Research Group, Faculty of Aerospace Engineering, TU Delft, Delft, The Netherlands
[2]Department of Mechanical Engineering, Ben-Gurion University of Negev, Beersheva, Israel

**Correspondence:** Vinit V. Dighe (V.V.Dighe@tudelft.nl)

**Abstract.** Multi-element ducts are used to improve the aerodynamic performance of ducted wind turbines (DWTs). Steady-state, two-dimensional computational fluid dynamics (CFD) simulations are performed for a multi-element duct geometry, consisting of a duct and a flap; goal is to evaluate the effects on the aerodynamic performance of the radial gap length and the deflection angle of the flap. Solutions from inviscid and viscous flow calculations are compared. It is found that increasing the radial gap length results in an augmentation of the total thrust generated by the DWT, whereas a larger deflection angle has an opposite effect. A reasonable to good agreement is seen between the inviscid and viscous flow calculations, except for multi-element duct configurations characterized by large flap deflection angles. The viscous effects become stronger at large flap deflection angles, and the inviscid calculations are incapable to take into account this phenomenon.

**1 Introduction**

Ducted wind turbines (DWTs) represent an interesting technological solution for increasing the energy extraction with respect to conventional horizontal axis wind turbines (HAWTs) for a given rotor radius and free stream velocity (de Vries , 1979). DWTs are constituted of a rotor and a duct; the role of the latter is to increase the mass flow rate through the rotor relative to a similar rotor operating in the open atmosphere, thereby increasing the generated power. There are more than one explanations for how this occurs. One explanation, as stated by van Bussel (2007), is that the duct forces an expansion of flow downstream of the turbine beyond what is attainable for a bare wind turbine. This provides a reduced pressure on the downstream of the turbine, and thereby increasing the total mass flow through the turbine. A second explanation, as argued by de Vries (1979), is that if the sectional lift force of the duct is directed towards the turbine plane, then the associated circulation of the duct induces an increased mass flow through the 
[revised manuscript text omitted]